# Channel Matters: Estimating Channel Influence for Multivariate Time Series

**Muyao Wang**[1,2]
muyaowang@stu.xidian.edu.cn

**Zeke Xie**[3]*
zekexie@hkust-gz.edu.cn

**Bo Chen**[1,2]*
bchen@mail.xidian.edu.cn

**Hongwei Liu**[1,2]
hwliu@xidian.edu.cn

**James Kwok**[4]
jamesk@cse.ust.hk

[1]Intitute of Information Sensing  [2]School of Electronic Engineering, Xidian University
[3]Hong Kong University of Science and Technology (Guangzhou)
[4]Hong Kong University of Science and Technology

## Abstract

The influence function serves as an efficient post-hoc interpretability tool that quantifies the impact of training data modifications on model parameters, enabling enhanced model performance, improved generalization, and interpretability insights without the need for expensive retraining processes. Recently, **M**ultivariate **T**ime **S**eries (MTS) analysis has become an important yet challenging task, attracting significant attention. While channel extremely matters to MTS tasks, channel-centric methods are still largely under-explored for MTS. Particularly, no previous work studied the effects of channel information of MTS in order to explore counterfactual effects between these channels and model performance. To fill this gap, we propose a novel Channel-wise Influence (ChInf) method that is the first to estimate the influence of different channels in MTS. Based on ChInf, we naturally derived two channel-wise algorithms by incorporating ChInf into classic MTS tasks. Extensive experiments demonstrate the effectiveness of ChInf and ChInf-based methods in critical MTS analysis tasks, such as MTS anomaly detection and MTS data pruning. Specifically, our ChInf-based methods rank top-1 among all methods for comparison, while previous influence functions do not perform well on MTS anomaly detection tasks and MTS data pruning problem. This fully supports the superiority and necessity of ChInf. Code is available at `https://github.com/flare200020/Chinf`.

## 1 Introduction

Multivariate time series (MTS) plays an important role in a wide variety of domains, including internet services [1] , industrial devices [2, 3] , health care [4, 5], finance [6, 7] , and so on. Thus, MTS modeling is crucial across a wide array of applications, including disease forecasting, traffic forecasting, anomaly detection, and action recognition. In recent years, researchers have focused on deep learning-based MTS analysis methods [8, 9, 10, 11, 12, 13, 14, 15]. Due to the large number of different channels in MTS, numerous studies aim to analyze the importance of these channels [12, 16, 17, 18]. Some of them concentrate on using graph or attention structure to capture the channel dependencies [12, 19], while some of them try to use Channel Independence to enhance the generalization ability on different channels of MTS model [17, 20]. While these methods try to

---

*Corresponding authors.

39th Conference on Neural Information Processing Systems (NeurIPS 2025).

fully utilize channel information, they fail to quantify the specific influence of each channel on model performance.

To quantitatively understand and improve MTS from a data-centric perspective, we report that influence functions may serve as a useful tool for MTS tasks. The influence function [21] is proposed to study the counterfactual effect between training data and model performance, which has been widely used in computer vision and natural language processing tasks, achieving promising results [22, 23, 24, 25, 26, 27]. However, in MTS, different channels exhibit complex correlations, making it particularly important to explore counterfactual effects between these channels and model performance. Considering that, it is essential to develop an appropriate influence function for MTS. It would provide extensions like increasing model performance, improving model generalization, and offering interpretability of the interactions between data channels and MTS models.

To the best of our knowledge, the influence of MTS in deep learning has not been well studied. It is nontrivial to apply the original influence function in [21] to this scenario, since different channels of MTS usually include different kinds of information and have various relationships [28, 12], which are important to MTS analysis. While recent work Timeinf [29] try to address temporal dependencies in time series, it overlooks the importance of channel-wise information and fails to achieve empirical success in practical MTS tasks. In principle, the previous influence functions can not distinguish the influence of different channels in MTS because they are designed for a whole data sample, according to their definitions. In practice, we also observe that the previous influence functions do not support anomaly detection effectively and perform not very well on MTS data pruning task, while they may perform well on computer vision and natural language process tasks [22, 24]. Thus, how to estimate the influence of different channels in MTS remains a critical problem.

To better understand and address the data-centric issues in MTS, we made three main contributions:

First, we propose a novel Channel-wise Influence (ChInf) method that is the first to be able to characterize the data influence of different channels for MTS.

Second, based on ChInf, we naturally derived two channel-wise enhancement algorithms for MTS anomaly detection and MTS data pruning tasks. This shows that ChInf can be easily applied to practical MTS tasks and methods.

Third, extensive experiments on various benchmark datasets demonstrate the superiority of ChInf and ChInf-based methods on the MTS anomaly detection and pruning tasks. Specifically, our ChInf-based methods rank top-1 among all methods, while previous influence functions do not perform well on some classic MTS anomaly detection tasks and MTS data pruning problem.

## 2 Related Work

In this section, we discuss related work.

### 2.1 Influence Functions

Influence functions estimate the effect of a given training example, $z'$, on a test example, $z$, for a pre-trained model. Specifically, the influence function estimates how removing a training example $z'$ affects the loss on a test example $z$. [21] derived the aforementioned influence to be $I(z', z) := \nabla_{\theta} L(z'; \theta)^{\top} H_{\theta}^{-1} \nabla_{\theta} L(z; \theta)$, where $H_{\theta}$ is the loss Hessian for the pre-trained model: $H_{\theta} := 1/n \sum_{i=1}^{n} \nabla_{\theta}^2 L(z; \theta)$, evaluated at the pre-trained model's final parameter checkpoint. The loss Hessian is typically estimated with a random mini-batch of data. The main challenge in computing influence is that it is impractical to explicitly form $H_{\theta}$ unless the model is small, or if one only considers parameters in a few layers. TracIn [27] address this problem by utilizing a first-order gradient approximation: $\text{TracIn}(z', z) := \nabla_{\theta} L(z'; \theta)^{\top} \nabla_{\theta} L(z; \theta)$, which has been proved effectively in various tasks [24, 22, 23, 30, 31]. Recent work, Timeinf [29], proposed an influence function capable of handling inherent temporal dependencies to better detect anomalies, which is literally orthogonal to our method. Moreover, it also did not touch the channel-wise analysis and failed to identify the value of channels for MTS.

## 2.2 Multivariate Time Series

There are various types of MTS analysis tasks. We mainly focus on unsupervised anomaly detection and preliminarily explore the value of our method in MTS forecasting.

**MTS Anomaly detection:** MTS anomaly detection has been extensively studied, including complex deep learning models [32, 9, 19, 33]. These models are trained to forecast or reconstruct presumed normal system states and then deployed to detect anomalies in unseen test datasets. The anomaly score, defined as the magnitude of prediction or reconstruction errors, serves as an indicator of abnormality at each timestamp. However, [34] have demonstrated that these methods create an illusion of progress due to flaws in the datasets [35] and evaluation metrics [36], and they provide a more fair and reliable benchmark. Additionally, unlike model-centric approaches, data-centric methods, which assess anomalies through influence measures that quantify the sensitivity of the model to input perturbations. Representative examples include TimeInf[29] and our proposed ChInf.

**MTS Forecasting:** In MTS forecasting, many methods try to model the temporal dynamics and channel dependencies effectively. An important issue in MTS forecasting is how to better generalize to unseen channels with a limited number of channels [12]. This places high demands on the model architecture, as the model must capture representative information across different channels and effectively utilize this information. There are two typical state-of-the-art methods to achieve this. One is iTransformer [12], which uses attention mechanisms to capture channel correlations. The other is PatchTST [17], which enhances the model's generalization ability by sharing the same model parameters across different channels through a Channel-Independence strategy.

State-of-the-art methods in MTS forecasting and MTS anomaly detection aim to fully utilize channel-wise information. Unlike previous model-centric approaches, we propose a data-centric method that leverages channel-wise influence information to enhance training data analysis and address MTS downstream tasks.

## 3 Channel-wise Influence

In this section, we formulate channel-wise influence with theoretical analysis, a novel and promising tool for MTS.

The influence function [21] requires the inversion of a Hessian matrix, which is quadratic in the number of model parameters. Fortunately, the original influence function can be accelerated and approximated by TracIn [27] effectively, which decomposes the difference between the loss of the test point at the end of training versus at the beginning of training along the path taken by the training process. The specific definition can be derived as follows:

$$\text{TracIn}\left(\boldsymbol{z}', \boldsymbol{z}\right) = L\left(\boldsymbol{z}; \boldsymbol{\theta}\right) - L(\boldsymbol{z}; \boldsymbol{\theta}') \approx \eta \nabla_{\boldsymbol{\theta}} L\left(\boldsymbol{z}'; \boldsymbol{\theta}\right)^{\top} \nabla_{\boldsymbol{\theta}} L(\boldsymbol{z}; \boldsymbol{\theta}) \tag{1}$$

where $\boldsymbol{z}'$ is the studied training example, $\boldsymbol{z}$ is the testing example, $\boldsymbol{\theta}$ is the well-trained model parameter without $\boldsymbol{z}'$, $\boldsymbol{\theta}'$ is the updated parameter after training with $\boldsymbol{z}'$, $L(\cdot)$ is the loss function, and $\eta$ is the learning rate during the training process. This formulation approximates the influence of $\boldsymbol{z}'$ by the inner product of the gradients of the training and test losses, scaled by $\eta$ ."

However, in the MTS analysis, the data sample $\boldsymbol{z}, \boldsymbol{z}'$ are MTS, which means TracIn can only calculate the whole influence of all channels. In other words, it fails to characterize the difference between different channels. To fill this gap we derive a new ChInf, using a derivation method similar to TracIn. Thus, we obtain Theorem 3.1 which formulates the ChInf, and the proof can be found in Appendix B.

**Theorem 3.1. (Channel-wise Influence Function)** *Assuming the $\boldsymbol{c}_i', \boldsymbol{c}_j$ is the i-th channel and j-th channel from the data sample $\boldsymbol{z}', \boldsymbol{z}$ respectively, $\boldsymbol{\theta}$ is the well-trained parameter of the model without $\boldsymbol{z}', L(\cdot)$ is the loss function and $\eta$ is the learning rate during the training process. The channel-wise influence under the first-order approximation can be formulated as follows:*

$$\textit{TracIn}\left(\boldsymbol{z}', \boldsymbol{z}\right) = \sum_{i=1}^{N} \sum_{j=1}^{N} \eta \nabla_{\boldsymbol{\theta}} L\left(\boldsymbol{c}_i'; \boldsymbol{\theta}\right)^{\top} \nabla_{\boldsymbol{\theta}} L\left(\boldsymbol{c}_j; \boldsymbol{\theta}\right) \tag{2}$$

Given the result, we define a channel-wise influence matrix $\boldsymbol{M}_{CInf}$ as follows:

$$\boldsymbol{M}_{CInf} = [a_{i,j}]_{N \times N},$$

where each element $a_{i,j}$ is defined as:

$$a_{i,j} := \eta \nabla_{\boldsymbol{\theta}} L \left( c_i'; \boldsymbol{\theta} \right)^{\top} \nabla_{\boldsymbol{\theta}} L \left( c_j; \boldsymbol{\theta} \right).$$

According to the theorem 3.1, the TracIn can be treated as a sum of these elements in the channel-wise influence matrix $\boldsymbol{M}_{CInf}$, failing to utilize the channel-wise information in the matrix specifically. Thus, the final ChInf can be defined as follows:

$$\text{CIF} \left( \boldsymbol{c}_i', \boldsymbol{c}_j \right) := \eta \nabla_{\boldsymbol{\theta}} L \left( \boldsymbol{c}_i'; \boldsymbol{\theta} \right)^{\top} \nabla_{\boldsymbol{\theta}} L \left( \boldsymbol{c}_j; \boldsymbol{\theta} \right) \tag{3}$$

where $\boldsymbol{c}_i', \boldsymbol{c}_j$ is the i-th channel and j-th channel from the data sample $\boldsymbol{z}, \boldsymbol{z}'$ respectively. This ChInf describes the influence between different channels among MTS.

*Remark* 3.2. **(Characteristics of Channel-wise Influence Matrix)** The channel-wise influence matrix reflects the relationships between different channels in a specific model. Specifically, each element $a_{i,j}$ in the matrix $\boldsymbol{M}_{CInf}$ represents how much training with channel i helps reduce the loss for channel j, which means similar channels usually have high influence score. Each model has its unique channel influence matrix, reflecting the model's way of utilizing channel information in MTS. Therefore, we can use $\boldsymbol{M}_{CInf}$ for post-hoc interpretable analysis of the model.

# 4 Methodology and Application

In this section, we discuss how to incorporate ChInf into classic MTS tasks, including anomaly detection and forecasting, naturally resulting in two ChInf-based methods. As channel matters to MTS tasks, we proposed a novel channel pruning challenge, which aims at achieve satisfying performance with less data channels and computational costs.

## 4.1 Multivariate Time Series Anomaly Detection

**Problem Definition:** Defining the training MTS as $\boldsymbol{x} = \{\boldsymbol{x}_1, \boldsymbol{x}_2, ..., \boldsymbol{x}_T\}$, where $T$ is the duration of $\boldsymbol{x}$ and the observation at time $t$, $\boldsymbol{x}_t \in \mathbb{R}^N$, is a $N$ dimensional vector where $N$ denotes the number of channels, thus $\boldsymbol{x} \in \mathbb{R}^{T \times N}$. The training data only contains non-anomalous timestep. The test set, $\boldsymbol{x}' = \{\boldsymbol{x}_1', \boldsymbol{x}_2', ..., \boldsymbol{x}_T'\}$ contains both normal and anomalous timestamps and $\boldsymbol{y}' = [\boldsymbol{y}_1', \boldsymbol{y}_2', ..., \boldsymbol{y}_T'] \in \{0, 1\}$ represents their labels, where $\boldsymbol{y}_t' = 0$ denotes a normal and $\boldsymbol{y}_t' = 1$ an anomalous timestamp t. Then the task of anomaly detection is to select a function $f_{\boldsymbol{\theta}} : X \to R$ such that $f_{\boldsymbol{\theta}}(\boldsymbol{x}_t) = \boldsymbol{y}_t$ estimates the anomaly score. When it is larger than the threshold, the data is predicted anomaly.

**Relationship between self-influence and anomaly score:** According to the conclusion in Section 4.1 of [27], influence can be an effective way to detect the anomaly sample. Specifically, the idea is to measure self-influence, i.e., the influence of a training point on its own loss, i.e., the training point $\boldsymbol{z}'$ and the test point $\boldsymbol{z}$ in TracIn are identical. From an intuitive perspective, self-influence reflects how much a model can reduce the loss during testing by training on sample $\boldsymbol{z}'$ itself. Therefore, anomalous samples, due to their distribution being inconsistent with normal training data, tend to reduce more loss, resulting in a greater self-influence. Therefore, when we sort test examples by decreasing self-influence, an effective influence computation method would tend to rank anomaly samples at the beginning of the ranking.

**Apply in MTS anomaly detection:** Based on these premises, we propose to derive an anomaly score based on the ChInf in Section 3 for MTS. Consider a test sample $\boldsymbol{x}'$ for which we wish to assess whether it is an anomaly. We can compute the channel-wise influence matrix $\boldsymbol{M}_{CInf}$ at first and then get the diagonal elements of the $\boldsymbol{M}_{CInf}$ to indicate the anomaly score of each channel. Since, according to the Remark 3.2 and the nature of self-influence, the diagonal elements reflect the ChInf, it is an effective method to reflect the anomaly level of each channel. Consistent with previous anomaly detection methods, we use the maximum anomaly score across different channels as the anomaly score of MTS $\boldsymbol{x}'$ at time $t$ as:

$$\text{Score} \left( \boldsymbol{x}_t' \right) := \max_i (\eta \nabla_{\boldsymbol{\theta}} L \left( \boldsymbol{c}_i'; \boldsymbol{\theta} \right)^{\top} \nabla_{\boldsymbol{\theta}} L \left( \boldsymbol{c}_i'; \boldsymbol{\theta} \right)) \tag{4}$$

where $\boldsymbol{c}_i'$ is the i-th channel of the MTS sample $\boldsymbol{x}_t'$, $\boldsymbol{\theta}$ is the trained parameter of the model, and $\eta$ is the learning rate during the training process. To ensure a fair comparison, we adopt the same anomaly score normalization and threshold selection strategy as outlined in [34] for detecting anomalies. Details regarding this methodology can be found in Appendix C. The comprehensive process for MTS anomaly detection is further elaborated in Algorithm 1.

| **Algorithm 1** ChInf based anomaly detection | **Algorithm 2** ChInf based MTS channel pruning |
|---|---|
| **Require:** test dataset $\mathcal{D}_{test}$; a well-trained network $\boldsymbol{\theta}$; loss function $L(\cdot)$; threshold $h$ 
 empty anomaly score dictionary $\rightarrow$ ADscore[]; 
 empty prediction dictionary $\rightarrow$ ADPredict[] 
 **for** $\boldsymbol{x} \in \mathcal{D}_{test}$ **do** 
 $\quad ADscore\,[\boldsymbol{x}] = \max_i(\eta \nabla_{\boldsymbol{\theta}} L\,(\boldsymbol{c}'_i; \boldsymbol{\theta})^\top \nabla_{\boldsymbol{\theta}} L\,(\boldsymbol{c}'_i; \boldsymbol{\theta}))$ 
 **end for** 
 Normalize $ADScore[\cdot]$    // Score normalization. 
 **if** $ADscore\,[\boldsymbol{x}] > h$ **then** 
 $\quad ADPredict\,[\boldsymbol{x}] = 1$    // Anomaly sample. 
 **else** 
 $\quad ADPredict\,[\boldsymbol{x}] = 0$    // Normal sample. 
 **end if** 
 return detection result $ADPredict\,[\cdot]$. | **Require:** val dataset $\mathcal{D}_{val}$; a well-trained network $\boldsymbol{\theta}$; loss function $L(\cdot)$; sample interval $t$ 
 empty channel set $\hat{\mathcal{D}} \rightarrow \{\}$ ; empty channel score dictionary $\rightarrow$ CScore[] 
 **for** $\boldsymbol{x} \in \mathcal{D}_{val}$ **do** 
 $\quad$ **for** $\boldsymbol{c}_i \in \boldsymbol{x}$ **do** 
 $\quad\quad CScore[\boldsymbol{c}_i] \mathrel{+}= \eta \nabla_{\boldsymbol{\theta}} L(\boldsymbol{c}_i; \boldsymbol{\theta})^\top \nabla_{\boldsymbol{\theta}} L(\boldsymbol{c}_i; \boldsymbol{\theta})$ 
 $\quad$ **end for** 
 **end for** 
 Sort(CScore)    // Sort influence scores 
 **for** $i = 0$ to $N$ **do** 
 $\quad$ **if** $i \bmod t == 0$ **then** 
 $\quad\quad$ Add $\boldsymbol{c}_i$ to $\hat{\mathcal{D}}$ 
 $\quad$ **end if** 
 **end for** 
 return pruned channel set $\hat{\mathcal{D}}$. |

## 4.2 Multivariate Time Series Data Pruning

**Dataset Pruning :** Dataset pruning remains a critical research focus, particularly as demonstrated by [37]'s findings showing its potential to overcome scaling law limitations through strategic dataset reduction while maintaining model performance. While the technique has been successfully applied to image [23, 22] and text [38, 39] datasets, its implementation in MTS analysis remains underdeveloped, presenting both technical challenges and untapped opportunities for efficiency optimization.

**Motivation:** iTransformer [12] demonstrates that training on part of channels enables effective full-channel predictions in MTS, indicating that there exists redundancy between MTS channels. Considering the importance and the necessity of dataset pruning [37], the excellent performance of the influence function in dataset pruning tasks [23, 22] and the discovery of iTransformer, we propose a new algorithm suitable for MTS named channel pruning to achieve the purpose of dataset pruning. With the help of channel pruning, we can accurately identify the subset of channels that are most representative for the model's training without retraining the model, resulting in MTS dataset pruning.

**Goal of Channel Pruning:** Given an MTS $\boldsymbol{x} = \{\boldsymbol{c}_1, ..., \boldsymbol{c}_N\}, \boldsymbol{y} = \{\boldsymbol{c}'_1, ..., \boldsymbol{c}'_N\}$ containing N channels where $\boldsymbol{c}_i \in R^{T \times 1}$, $\boldsymbol{x}$ is the input space and $\boldsymbol{y}$ is the label space. Channel pruning aims to identify a set of representative channels from $\boldsymbol{x}$ as few as possible to reduce the training cost and find the relationship between model and channels. The identified representative subset, $\hat{\mathcal{D}} = \{\hat{\boldsymbol{c}}_1, ..., \hat{\boldsymbol{c}}_m\}$ and $\hat{\mathcal{D}} \subset \mathcal{D}$, where $\mathcal{D}$ is the full dataset, should have a maximal impact on the learned model, i.e. the test performances of the models learned on the training sets before and after pruning should be close, as described below:

$$\mathbb{E}_{\boldsymbol{c} \sim P(\mathcal{D})} L(\boldsymbol{c}, \theta) \simeq \mathbb{E}_{\boldsymbol{c} \sim P(\mathcal{D})} L\left(\boldsymbol{c}, \theta_{\hat{\mathcal{D}}}\right) \tag{5}$$

where $P(\mathcal{D})$ is the channel distribution, $L(\cdot)$ is the loss function, and $\boldsymbol{\theta}$ and $\boldsymbol{\theta}_{\hat{\mathcal{D}}}$ are the empirical risk minimizers on the training set $\mathcal{D}$ before and after pruning $\hat{\mathcal{D}}$, respectively, i.e., $\boldsymbol{\theta} = \arg\min_{\boldsymbol{\theta} \in \Theta} \frac{1}{N} \sum_{\boldsymbol{c}_i \in \mathcal{D}} L\,(\boldsymbol{c}_i, \boldsymbol{\theta})$ and $\boldsymbol{\theta}_{\hat{\mathcal{D}}} = \arg\min_{\boldsymbol{\theta} \in \Theta} \frac{1}{m} \sum_{\boldsymbol{c}_i \in \hat{\mathcal{D}}} L\,(\boldsymbol{c}_i, \boldsymbol{\theta})$.

**Apply in channel pruning:** Considering the channel-wise data pruning problem, our proposed ChInf method can effectively address this issue. According to the Remark 3.2, our approach can use $M_{CInf}$ to represent the characteristics of each channel by calculating the influence of different channels. Then, we use a concise approach to obtain a representative subset of channels. Specifically, we can rank the diagonal elements of $M_{CInf}$, i.e., the ChInf, and select the subset of channels at regular intervals for a certain model. Since similar channels have a similar self-influence, we can adopt regular sampling on the original channel set $\mathcal{D}$ based on the ChInf to acquire a representative subset of channels $\hat{\mathcal{D}}$ for a certain model and dataset, which is typically much smaller than the original dataset. The detailed process of channel pruning is shown in Algorithm 2. Consequently, we can train or fine-tune the model with a limited set of data efficiently. Additionally, it can serve as an explainable method to reflect the channel-modeling ability of different approaches. Specifically, the

smaller the size of the representative subset $\hat{\mathcal{D}}$ for a method, the fewer channels' information it uses for predictions, and vice versa. Thus, a good MTS modeling method should have a large size of $\hat{\mathcal{D}}$.

## 5   Empirical Analysis

In this section, we mainly discuss the performance of our method in MTS anomaly detection and explore the value and feasibility of our method in MTS data pruning tasks. All the datasets used in our experiments are real-world and open-source MTS datasets.

### 5.1   Mutivariate Time Series Anomaly Detection

#### 5.1.1   Baselines and Experimental Settings

We conduct model comparisons across five widely-used anomaly detection datasets: SMD[40], MSL [41], SMAP [41], SWaT [42], and WADI [19]. Given the point-adjustment evaluation metric is proved unreasonable [34, 36], we use the standard precision, recall and F1 score to measure the performance, aligning with [34]. Moreover, due to the flaws in the previous methods, [34] provides a more fair benchmark, including many simple but effective methods, such as GCN-LSTM, PCA ERROR and so on, labeled as **Simple baseline** in the Table 1. Thus, for a fair comparison, we use the same data preprocessing as described in [34] and use the results cited from their paper or reproduced with their code as strong baselines. Considering iTransformer [12] can capture the channel dependencies with attention block adaptively and the good performance of Timeinf in anomaly detection, we also add them as new baselines.

Table 1: Experimental results for SWaT, SMD, MSL, SMAP, and WADI datasets. The bold and underlined marks are the best and second-best value. F1: the standard F1 score; P: Precision; R: Recall. (For SMD, MSL, and SMAP, the Precision, Recall, and F1 scores are computed for each trace and then averaged to obtain the final results.) Higher values indicate better performance.

| Method | Datasets | | | | | | | | | | | | | | |
| --- | --- | --- | --- | --- | --- | --- | --- | --- | --- | --- | --- | --- | --- | --- | --- |
| | SWAT | | | SMD | | | SMAP | | | MSL | | | WADI | | |
| | F1 | P | R | F1 | P | R | F1 | P | R | F1 | P | R | F1 | P | R |
| DAGMM [43] | 77.0 | 99.1 | 63.0 | 43.5 | 56.4 | 49.7 | 33.3 | 39.5 | 56.0 | 38.4 | 40.1 | 59.6 | 27.9 | 99.3 | 16.2 |
| OmniAnomaly [40] | 77.3 | 99.0 | 63.4 | 41.5 | 56.6 | 46.4 | 35.1 | 37.2 | 62.5 | 38.7 | 40.7 | 61.5 | 28.1 | 100 | 16.3 |
| USAD [44] | 77.2 | 98.8 | 63.4 | 42.6 | 54.6 | 47.4 | 31.9 | 36.5 | 40.2 | 38.6 | 40.2 | 61.1 | 27.9 | 99.3 | 16.2 |
| GDN [19] | 81.0 | 98.7 | 68.6 | 52.6 | 59.7 | 56.5 | 42.9 | 48.2 | 63.1 | 44.2 | 38.6 | 62.4 | 34.7 | 64.3 | 23.7 |
| TranAD [9] | 80.0 | 99.0 | 67.1 | 45.7 | 57.9 | 48.1 | 35.8 | 37.8 | 52.5 | 38.1 | 40.1 | 59.7 | 34.0 | 29.3 | 40.4 |
| AnomalyTransformer [33] | 76.5 | 94.3 | 64.3 | 42.6 | 41.9 | 52.8 | 31.1 | 42.3 | 60.4 | 33.8 | 31.3 | 59.8 | 20.9 | 12.2 | 74.3 |
| PCA ERROR (Simple baseline) | 83.3 | 96.5 | 73.3 | 57.2 | 61.1 | 58.4 | 39.2 | 43.4 | 65.5 | 42.6 | 39.6 | 63.5 | 50.1 | 88.4 | 35.0 |
| 1-Layer MLP (Simple baseline) | 77.1 | 98.1 | 63.5 | 51.4 | 59.8 | 57.4 | 32.3 | 43.2 | 58.7 | 37.3 | 34.2 | 64.8 | 26.7 | 83.4 | 15.9 |
| Single block MLPMixer (Simple baseline) | 78.0 | 85.4 | 71.8 | 51.2 | 60.8 | 55.4 | 36.3 | 45.1 | 61.2 | 39.7 | 34.1 | 62.8 | 27.5 | 86.2 | 16.3 |
| Single Transformer block (Simple baseline) | 78.7 | 86.8 | 72.0 | 48.9 | 58.9 | 53.6 | 36.6 | 42.4 | 62.9 | 40.2 | 42.7 | 56.9 | 28.9 | 90.8 | 17.2 |
| 1-Layer GCN-LSTM (Simple baseline) | 82.9 | 98.2 | 71.8 | 55.0 | 62.7 | 59.9 | 42.6 | 46.9 | 61.6 | 46.3 | 45.6 | 58.2 | 43.9 | 74.4 | 31.1 |
| Using ChInf (Ours) | 82.9 | 98.0 | 71.8 | 58.8 | 63.5 | 62.2 | **48.0** | 54.3 | 59.6 | **47.1** | 41.1 | 67.6 | 47.2 | 54.5 | 41.6 |
| Inverted Transformer [12] | 83.7 | 96.3 | 74.1 | 55.9 | 65.0 | 57.0 | 39.6 | 49.7 | 60.8 | 45.5 | 44.8 | 66.6 | 48.8 | 64.2 | 39.4 |
| Using ChInf (Ours) | **84.0** | 96.4 | 74.4 | **59.1** | 63.6 | 63.8 | 46.3 | 52.9 | 61.3 | 46.1 | 41.9 | 68.4 | **50.5** | 58.7 | 44.2 |
| Timeinf [29] | 79.0 | 91.7 | 69.4 | 54.1 | 64.4 | 61.2 | 35.1 | 45.6 | 65.9 | 39.7 | 41.7 | 64.5 | 27.1 | 89.4 | 15.6 |

#### 5.1.2   Main Results

In this experiment, we compare our ChInf method with other model-centric methods. Apparently, Table 1 showcases the superior performance of our method, achieving the highest F1 score among the previous state-of-the-art (SOTA) methods. The above results demonstrate the effectiveness of our ChInf and ChInf-based anomaly detection method. Specifically, the use of model gradient information in self-influence highlights that the gradient information across different layers of the model enables the identification of anomalous information, contributing to good performance in anomaly detection.

#### 5.1.3   Additional Analysis

In this section, we conduct several experiments to validate the effectiveness of the ChInf and explore its characteristics.

**Ablation Study:**   In our method, the most important part is the design of ChInf and replacing the reconstructed or predicted error with our ChInf to detect the anomalies. We conduct ablation studies

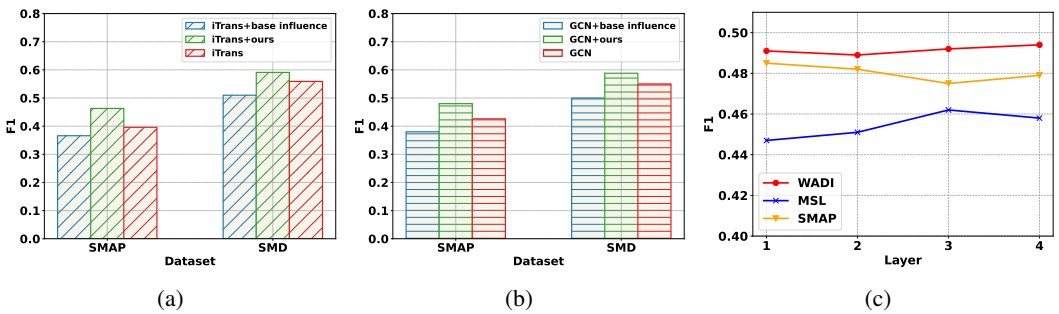

Figure 1: (a)-(b): The ablation study of ChInf for iTransformer and GCN-LSTM on SMAP and SMD dataset. Our ChInf can enhance MTS performance, while the conventional influence fails. (c): The relationship between the number of parameters used to calculate influence and the anomaly detection performance on different datasets.

on different datasets and models. Fig 1a and Fig 1b show that the ChInf is better than the original TracIn [27] and the original TracIn is worse than the reconstructed error. It is because that the original TracIn fails to distinguish which channel is abnormal more specifically. Additionally, both figures demonstrate that our method achieves strong performance across different models, underscoring the effectiveness and generalization capability of our data-centric approach. Given the superiority of ChInf over the TracIn, the design of a dedicated ChInf becomes essential.

**Trade-off Analysis:** According to the formula Eq. 3, we need to compute the model's gradient. Considering computational efficiency, we use the gradients of a subset of the model's parameters to calculate influence. Therefore, we tested the relationship between the number of parameters used and the anomaly detection performance, with the results shown in Fig. 1c. Specifically, we use the GCN-LSTM model as an example, which has an MLP decoder containing two linear layers, each with weight and bias parameters. Therefore, we can use these four layers to test the effect of the number of parameters used. The results in Fig. 1c indicate that our method is not sensitive to the choice of parameters. Hence, using only the gradients of the last layer of the network is sufficient to achieve excellent performance in approximating the influence, which aligns with the conclusions in [22, 24, 27]. In addition, we also test the inference time to show the complexity of our method in Appendix D.5.

**Model architecture generalization:** To further demonstrate the generalizability of our method, we applied our ChInf to various model architectures and presented the results in Table 2, bold marks indicate the best results. As clearly shown in the table, our method consistently exhibited superior performance across different model architectures. Therefore, we can conclude that our method is suitable for different types of models, proving that it is a qualified data-centric approach. Full results of the analysis are in Table 8 (Appendix).

Table 2: Generalization ability of our method with different model architectures. Bold indicates best performance.

| Method | | MLP-Mixer | | | Transformer | | |
|---|---|---|---|---|---|---|---|
| Dataset | | F1 | P | R | F1 | P | R |
| SMD | Recon | 51.2 | 60.8 | 55.4 | 48.9 | 58.9 | 53.6 |
| | ChInf | **55.5** | 64.8 | 58.3 | **52.1** | 62.9 | 58.2 |
| SMAP | Recon | 36.3 | 45.1 | 61.2 | 36.6 | 42.4 | 62.9 |
| | ChInf | **48.0** | 57.5 | 58.9 | **48.5** | 54.1 | 64.6 |
| MSL | Recon | 39.7 | 34.1 | 62.8 | 40.2 | 42.7 | 56.9 |
| | ChInf | **46.2** | 44.6 | 57.1 | **47.7** | 42.8 | 64.9 |

**Visualization of Anomaly Score:** To highlight the differences between our ChInf method and traditional reconstruction-based methods, We visualized the anomaly scores obtained from the SMAP dataset. Apparently, as indicated by the red box in Fig. 4 in the Appendix, the reconstruction error fails to fully capture the anomalies, making it difficult to distinguish some normal samples from the anomalies, as their anomaly scores are similar to the threshold. The results show that our method can detect true anomalies more accurately compared to reconstruction-based methods, demonstrating the advantage of ChInf.

Table 3: Channel pruning experimental results for Electricity, Solar Energy, and Traffic datasets. We use the MSE metric to reflect the performance of different methods. The bold marks are the best. The predicted length is 96. The red markers indicate the proportion of channels that can achieve a satisfying performance with ChInf-based selection.

| Dataset | | ECL | | | | | Solar | | | | | Traffic | | | | |
|---|---|---|---|---|---|---|---|---|---|---|---|---|---|---|---|---|
| Proportion of variables retained | | 5% | 10% | 15% | 20% | 50% | 5% | 10% | 15% | 20% | 50% | 5% | 10% | 15% | 20% | 30% |
| iTransformer | Continuous selection | 0.208 | 0.188 | 0.181 | 0.178 | 0.176 | 0.241 | 0.228 | 0.225 | 0.224 | 0.215 | 0.470 | 0.437 | 0.409 | 0.406 | 0.404 |
| | LIME selection | 0.195 | 0.185 | 0.174 | 0.172 | 0.151 | 0.244 | 0.235 | 0.231 | **0.214** | 0.211 | 0.452 | 0.436 | 0.424 | 0.414 | 0.408 |
| | SHAP selection | 0.193 | **0.173** | 0.171 | 0.166 | 0.151 | 0.248 | 0.228 | 0.220 | 0.217 | 0.212 | 0.449 | 0.423 | 0.416 | 0.410 | 0.405 |
| | NFS selection | 0.201 | 0.185 | 0.180 | 0.177 | 0.167 | 0.260 | 0.248 | 0.227 | 0.222 | 0.214 | 0.428 | 0.408 | 0.402 | 0.399 | 0.397 |
| | Influence selection | **0.187** | 0.174 | **0.170** | **0.165** | **0.150** | **0.229** | **0.224** | **0.220** | 0.219 | **0.210** | **0.419** | **0.405** | **0.398** | **0.397** | **0.395** |
| | Full variates | | 0.148 | | | | | 0.206 | | | | | 0.395 | | | |
| Proportion of variables retained | | 5% | 10% | 15% | 20% | 45% | 5% | 10% | 15% | 20% | 20% | 5% | 10% | 15% | 20% | 20% |
| PatchTST | Continuous selection | 0.304 | 0.222 | 0.206 | 0.202 | 0.203 | 0.250 | 0.244 | 0.240 | 0.230 | 0.230 | 0.501 | 0.478 | 0.474 | 0.476 | 0.476 |
| | LIME selection | 0.210 | 0.196 | 0.192 | 0.190 | 0.180 | 0.245 | 0.230 | 0.228 | 0.226 | 0.226 | 0.493 | 0.478 | 0.467 | 0.461 | 0.461 |
| | SHAP selection | 0.215 | 0.211 | 0.200 | 0.195 | 0.186 | 0.239 | 0.236 | 0.229 | 0.227 | 0.227 | 0.505 | 0.487 | 0.482 | 0.480 | 0.480 |
| | NFS selection | 0.222 | 0.199 | 0.196 | 0.192 | 0.186 | 0.240 | 0.231 | 0.228 | 0.226 | 0.226 | 0.494 | 0.474 | 0.465 | 0.460 | 0.460 |
| | Influence selection | **0.205** | **0.191** | **0.190** | **0.186** | **0.176** | **0.228** | **0.226** | **0.226** | **0.223** | **0.223** | **0.483** | **0.470** | **0.456** | **0.452** | **0.452** |
| | Full variates | | 0.176 | | | | | 0.223 | | | | | 0.452 | | | |

## 5.2 Multivariate Time Series Data Pruning

### 5.2.1 Channel Pruning Experiment

**Set Up:** To demonstrate the effectiveness of our method, we designed a channel pruning experiment similar to dataset pruning [22] in MTS forecasting task. In this experiment, we used three benchmark datasets with a large number of channels for testing: Electricity with 321 channels, Solar-Energy with 137 channels, and Traffic with 821 channels. According to Eq.5, the specific aim of the experiment was to determine how to retain only $M\%$ of the channels while maximizing the model's generalization ability across all channels. In addition to our proposed method, we compared it with some naive baseline methods, including training with the first $M\%$ of the channels, Lime selection [45], SHAP selection [46], and NFS channel selection method [47]. $M$ is changed to demonstrate the channel-pruning ability of these methods.

**Results Analysis:** The bold mark results in the Table.3 indicate that, when retaining the same proportion of channels, our method outperforms other methods in most cases. Besides, the red mark results in the table also show that our method can maintain the original prediction performance while using no more than half of the channels, outperforming other baseline methods. Although LIME and SHAP can achieve comparable performance to our method under certain settings, our approach offers several distinct advantages. These include higher computational efficiency, the ability to perform not only post-hoc analysis but also anomaly detection, and the capability to analyze inter-channel correlations. For a detailed discussion, please refer to Appendix D.5. To further prove the effectiveness of our method, we test our method on more datasets and models in the Appendix D.3. Additionally, considering our selection strategy is different from conventional wisdom, such as selecting the most influential samples, we add comparison experiments in Appendix D.2 to prove the effectiveness of our method.

**Interpretable analysis:** In addition to the superior performance shown in the Table 3, the size of the representative subset $\hat{\mathcal{D}}$, mentioned in Table 4, can be served as a metric to measure the performance of an MTS model. Specifically, it is evident that iTransformer [12] has a larger representative subset than PatchTST [17], which means iTransformer can utilize different kinds of channel information more effectively. This explains why iTransformer has a better predictive performance.

Table 4: The size of representative sets of different models on different datasets, summarized from Table 3.

| Model | ECL | Solar | Traffic |
|---|---|---|---|
| iTransformer | **50%** | **50%** | **30%** |
| PatchTST | 45% | 20% | 20% |

**Outlook:** Based on the results in Table 4, the iTransformer has a larger representative subset, we believe that in addition to using the ChInf for channel pruning to improve the efficiency of model training and fine-tuning, another important application is its use as a post-hoc interpretable method to evaluate a model's quality. As our experimental results demonstrate, a good model should be able to fully utilize the information between different channels. Therefore, to achieve the original performance, such a method would require retaining a higher proportion of channels.

### 5.2.2 Case Study and Interpretability Analysis

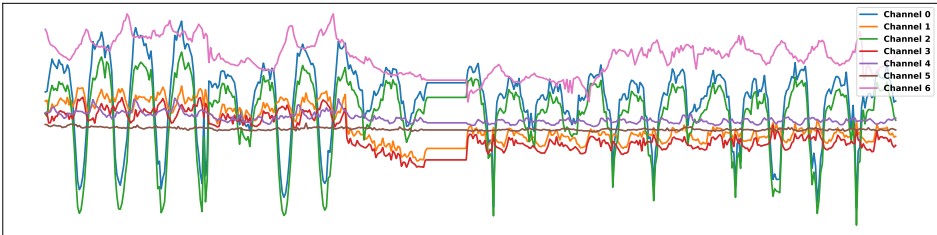

Figure 2: Visual illustration of different channels on ETTh1 dataset.

To further investigate whether the channels ranked by ChInf correspond to domain-critical variables, we conduct a qualitative case study on the *ETTh1* dataset, which contains seven channels with explicit physical meanings: HUFL(0) (High Utilization Factor Load), HULL(1) (High Utilization Low Load), MUFL(2) (Medium Utilization Factor Load), MULL(3) (Medium Utilization Low Load), LUFL(4) (Low Utilization Factor Load), LULL(5) (Low Utilization Low Load), and OT(6) (Oil Temperature).

According to our experiment, ChInf-based pruning yields forecasting performance comparable to using all seven channels. Specifically, the top-ranked channels selected by ChInf are HULL (1), MUFL (2), LUFL (4), and OT (6).

We further analyze the cross-channel dependencies using visualization-based analysis as shown in Fig 2. The temporal patterns indicate that:

- HUFL (0) and MUFL (2) exhibit similar temporal dynamics.
- HULL (1) and MULL (3) exhibit similar temporal dynamics.
- LUFL (4) and LULL (5) show relatively weak similarity.
- OT (6) is largely independent of the other load-related channels.

These results demonstrate that ChInf successfully identifies representative and diverse channels that capture distinct temporal behaviors while preserving cross-channel coverage. The selected subset (1, 2, 4, 6) aligns well with known domain structure, indicating that the ChInf rankings reflect meaningful channel importance consistent with real-world physical interpretation.

### 5.2.3 Sample Pruning versus Channel Pruning

**Set up:** To further demonstrate the superiority and the necessity of channel pruning, we conducted a comparative experiment between sample-wise data pruning and channel pruning. Specifically, we reduced the training data size using two pruning strategies: for sample pruning, we applied MoSo [23], an effective sample pruning approach, alongside random sample pruning, which involved randomly selecting data samples. For channel pruning, we utilized our ChInf. We compared each pruning method at the same remaining ratio. For example, when the horizontal axis in Fig. 3 indicates that 50% is retained, it means the size of the entire dataset is reduced to half of its original size. In the case of sample pruning, half of the training samples will be discarded; whereas in channel pruning, half of the channels will be discarded.

**Result Analysis:** As shown in Fig. 3, our channel pruning method achieved better performance while retaining the same proportion of data on all settings. This suggests that channel pruning is a more suitable method for reducing MTS data size. Additionally, we previously highlighted the value of channel pruning as a post-hoc method for analyzing MTS models. Therefore, we believe that channel pruning holds greater exploratory value in MTS.

Furthermore, we found that the performance of the MoSo-based pruning method was not as effective as that of the random pruning method. We believe this may be due to the traditional influence method underlying MoSo, which assumes that each data sample is calculated in isolation. However, the samples in time series forecasting usually have strong dependencies, thus resulting in the failure of the MoSo method. Therefore, we consider designing an effective dataset pruning method specifically for time series forecasting to be a noteworthy problem.

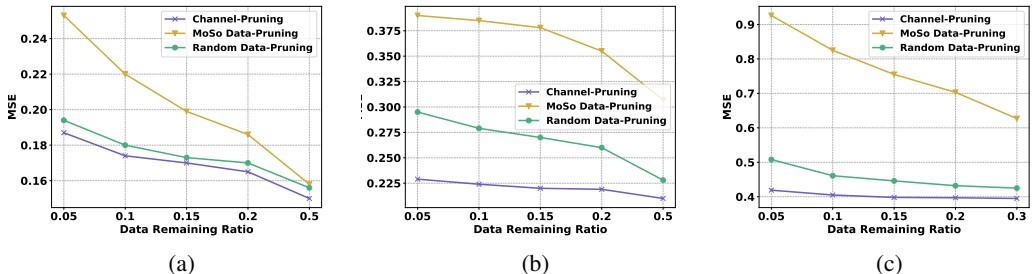

Figure 3: (a)-(c): The comparison experiment between sample pruning and channel pruning on three datasets. From left to right are the Electricity dataset, the Solar Energy dataset, and the Traffic dataset. The evaluation metric used is mean squared error (MSE), with lower values indicating better performance. The horizontal axis represents the remaining ratio of the dataset.

# 6  Conclusion and Discussion

Channel matters to MTS tasks. However, channel-centric methods are still largely under-explored for MTS. In this paper, we introduce a novel ChInf method, which is the first capable of estimating the influence of individual channels in MTS. Unlike traditional model-centric approaches, our data-centric method demonstrates superior performance in anomaly detection and channel pruning and address the limitations of existing influence functions for MTS. Extensive experiments on real-world datasets validate the effectiveness and versatility of our approach for various MTS analysis tasks. We report that channel pruning, a novel dataset pruning method, can prune more redundant MTS data than existing sample-wise data pruning methods without notable loss of performance. We also revealed two usually overlooked issues in MTS forecasting: the issue of insufficient data utilization by current MTS models and the potential redundancy in existing datasets. In summary, our method not only provides a powerful data-centric tool for MTS analysis but also offers valuable insights into the characteristics of channel information in MTS. Our work paves the way for future development of MTS methods from a data-centric perspective.

**Limitation:** While we have successfully applied our method to two fundamental MTS tasks and demonstrated its effectiveness over previous influence functions and model-centric methods, our empirical analysis mainly focused on MTS anomaly detection and MTS data pruning tasks. There remains a vast landscape of MTS-related tasks that are yet to be explored and understood.

**Targets Anomalies of ChInf:**  Inspired by previous research [48, 49, 50], we discuss the types of anomalies that our method is designed to target. ChInf demonstrates particular effectiveness in detecting localized, channel-wise anomalies, where individual channels deviate from their normal interactions with others. In such cases, the model parameters exhibit abnormal sensitivity to specific inputs—an effect naturally captured by self-influence estimation. This property enables ChInf to identify "broken channels" or malfunctioning sensors, aligning with prior findings emphasizing the importance of modeling inter-channel dependencies. Conversely, ChInf may be less effective for globally synchronized anomalies [51, 52] that occur uniformly across all channels, as these do not induce distinct channel-wise influence patterns. Regarding the datasets used in our experiments, many contain subset-sensor failures or transient spikes, which make them particularly suitable for evaluating ChInf's strengths.

**Future Directions:** Looking ahead, a primary focus of our research will be the further application of the ChInf. We believe that delving deeper into this area will yield valuable insights and contribute significantly to advancing the field. Our findings also highlight two usually overlooked issues in MTS forecasting: the issue of insufficient data utilization by current MTS models and the potential redundancy in existing datasets. These insights also suggest two wide-scope future research directions: improving model efficiency in data utilization and enhancing dataset quality.

## Acknowledgments

This work was supported in part by the National Natural Science Foundation of China under Grant 62576266 and U21B2006; in part by the Fundamental Research Funds for the Central Universities QTZX24003 and QTZX23018; in part by the 111 Project under Grant B18039. This research was supported in part by the Research Grants Council of the Hong Kong Special Administrative Region (Grant 16202523 and HKU C7004-22G). This work was supported by Guangdong Provincial Key Lab of Integrated Communication, Sensing and Computation for Ubiquitous Internet of Things (No.2023B1212010007).

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

## A  Broader Impact

Our model is well-suited for multivariate time series analysis tasks, offering practical and positive impacts across various domains, including disease forecasting, traffic prediction, internet services, content delivery networks, wearable devices, and action recognition. However, we emphatically discourage its application in activities related to financial crimes or any other endeavors that could lead to negative societal consequences.

## B  Proof

*Proof.* The proof of Theorem 3.1:

$$
\begin{aligned}
\text{TracIn}\,(\boldsymbol{z}', \boldsymbol{z}) &= L\,(\boldsymbol{z}; \boldsymbol{\theta}) - L(\boldsymbol{z}; \boldsymbol{\theta}') \\
&= \sum_{i=1}^{N} L\,(\boldsymbol{c}_i; \boldsymbol{\theta}) - \sum_{j=1}^{N} L(\boldsymbol{c}_j; \boldsymbol{\theta}') \\
&= \sum_{i=1}^{N} \left( \nabla L\,(\boldsymbol{c}_i; \boldsymbol{\theta}) \cdot (\boldsymbol{\theta}' - \boldsymbol{\theta}) + O\left( \|\boldsymbol{\theta}' - \boldsymbol{\theta}\|^2 \right) \right) \\
&\approx \sum_{i=1}^{N} \nabla L\,(\boldsymbol{c}_i'; \boldsymbol{\theta})\, \eta \nabla L\,(\boldsymbol{z}; \boldsymbol{\theta}) \\
&= \sum_{i=1}^{N} \sum_{j=1}^{N} \eta \nabla L\,(\boldsymbol{c}_i'; \boldsymbol{\theta})\, \nabla L\,(\boldsymbol{c}_j; \boldsymbol{\theta})
\end{aligned}
\tag{6}
$$

where the first equation is the original definition of TracIn; we rectify the equation and derive the second equation, indicating the sum of the loss of each channel. The third equation is calculated by the first approximation of the loss function and then we replace $(\boldsymbol{\theta}' - \boldsymbol{\theta})$ with $\eta \nabla L\,(\boldsymbol{z}'; \boldsymbol{\theta})$. Therefore, we can derive the final equation which demonstrates the original TracIn at the channel-wise level.

The proof is complete.

$\square$

## C  Details of Experiments

### C.1  Dataset details

**Anomaly detection:**

Since SMD, SMAP, and MSL datasets contain traces with various lengths in both the training and test sets, we report the average length of traces and the average number of anomalies among all traces per dataset. The detailed information of the datasets can be found in Table. 5.

Table 5: The detailed dataset information.

| Dataset | Sensors(traces) | Train | Test | Anomalies |
|---------|-----------------|-------|------|-----------|
| SWaT | 51 | 47520 | 44991 | 4589(12.2%) |
| WADI | 127 | 118750 | 17280 | 1633(9.45%) |
| SMD | 38(28) | 25300 | 25300 | 1050(4.21%) |
| SMAP | 25(54) | 2555 | 8070 | 1034(12.42%) |
| MSL | 55(27) | 2159 | 2730 | 286(11.97%) |

**Time series forecasting:**

The detailed information of these datasets can be found in the Table 6.

Table 6: The detailed dataset information.

| Dataset | Dim | Prediction Length | Datasize | Frequency |
|---------|-----|-------------------|----------|-----------|
| Electricity | 321 | 96 | (18317, 2633, 5261) | Hourly |
| Solar-Energy | 137 | 96 | (36601, 5161, 10417) | 10min |
| Traffic | 862 | 96 | (12185, 1757, 3509) | Hourly |

## C.2 Training Details

All experiments were implemented using PyTorch and conducted on a single NVIDIA GeForce RTX 3090 24GB GPU.

**For anomaly detection:** Models were trained using the SGD optimizer with Mean Squared Error (MSE) loss. For both of them, when trained in reconstructing mode, we used a time window of size 10.

**For channel pruning:** Models were trained using the Adam optimizer with Mean Squared Error (MSE) loss. The input length is 96 and the predicted length is 96.

**Anomaly Score Normalization**   Anomaly detection methods for multivariate datasets often employ normalization and smoothing techniques to address abrupt changes in prediction scores that are not accurately predicted. In this paper, we mainly use two normalization methods, mean-standard deviation and median-IQR, which aligns with [34]. The details are as follows:

$$s_i = \frac{S_i - \widetilde{\mu}_i}{\tilde{\sigma}_i} \tag{7}$$

**For median-IQR:** The $\widetilde{\mu}$ and $\tilde{\sigma}$ are the median and inter-quartile range (IQR2) across time ticks of the anomaly score values respectively.

**For mean-standard deviation:** The $\widetilde{\mu}$ and $\tilde{\sigma}$ are the mean and standard across time ticks of the anomaly score values respectively.

For a fair comparison, we select the best results of the two normalization methods as the final result, which aligns with [34].

**Threshold Selection**   Typically, the threshold which yields the best F1 score on the training or validation data is selected. This selection strategy aligns with [34], for a fair comparison.

## D   Additional Empirical Analysis

### D.1   Visualization of anomaly detection

To highlight the differences between our ChInf method and traditional reconstruction-based methods, We visualized the anomaly scores obtained from the SMAP dataset. Apparently, as indicated by the red box in Fig. 4, the reconstruction error fails to fully capture the anomalies, making it difficult to distinguish some normal samples from the anomalies, as their anomaly scores are similar to the threshold. The results show that our method can detect true anomalies more accurately compared to reconstruction-based methods, demonstrating the advantage of ChInf.

### D.2   Utilization of ChInf

We conducted new experiments comparing different selection strategies based on ChInf. The results, shown in the table 7, indicate that our equidistant sampling approach is more effective than selecting the most influential samples. This is because it covers a broader range of channels, allowing the model to learn more general time-series patterns during training.

### D.3   Generalization results

To demonstrate the generalizability of our method, we applied our ChInf to various model architectures and presented the results in the following table 8. As clearly shown in the table, our method

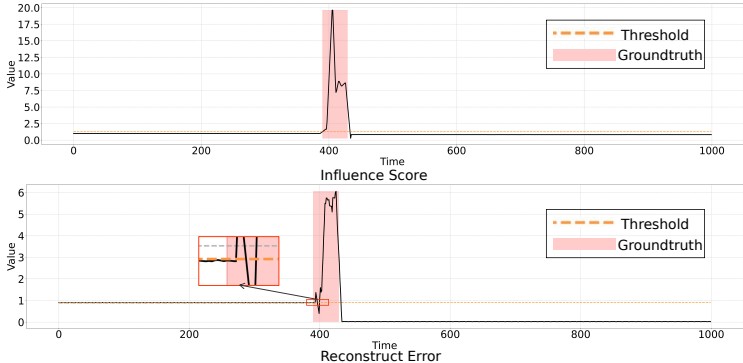

Figure 4: Visual illustration of the anomaly score of different methods on SMAP dataset.

Table 7: Variate generalization experimental results for Electricity, Solar Energy, and Traffic datasets. We use the MSE metric to reflect the performance of different methods. The bold marks are the best. The predicted length is 96. The red markers indicate the proportion of channels that need to be retained to achieve the original prediction performance.

| Dataset | | ECL | | | | | Solar | | | | | Traffic | | | | |
|---|---|---|---|---|---|---|---|---|---|---|---|---|---|---|---|---|
| Proportion of variables retained | | 5% | 10% | 15% | 20% | 50% | 5% | 10% | 15% | 20% | 50% | 5% | 10% | 15% | 20% | 30% |
| iTransformer | Most self-influence sample | 0.360 | 0.224 | 0.181 | 0.176 | 0.160 | 0.351 | 0.241 | 0.237 | 0.236 | 0.220 | 0.461 | 0.421 | 0.407 | 0.401 | 0.399 |
| | Ours | **0.187** | **0.174** | **0.170** | **0.165** | **0.150** | **0.229** | **0.224** | **0.220** | **0.219** | **0.210** | **0.419** | **0.405** | **0.398** | **0.397** | **0.395** |
| | Full variates | | | 0.148 | | | | | 0.206 | | | | | 0.395 | | |

consistently exhibited superior performance across different model architectures. Therefore, we can conclude that our method is suitable for different types of models, proving that it is a qualified data-centric approach.

Table 8: Full results of the generalization ability experiment.

| Method | | 1-Layer MLP | | | Single block MLPMixer | | | Single Transformer block | | |
|---|---|---|---|---|---|---|---|---|---|---|
| | Dataset | F1 | P | R | F1 | P | R | F1 | P | R |
| SMD | Reconstruct Error | 51.4 | 59.8 | 57.4 | 51.2 | 60.8 | 55.4 | 48.9 | 58.9 | 53.6 |
| | ChInf | **55.9** | 63.1 | 60.6 | **55.5** | 64.8 | 58.3 | **52.1** | 62.9 | 58.2 |
| SMAP | Reconstruct Error | 32.3 | 43.2 | 58.7 | 36.3 | 45.1 | 61.2 | 36.6 | 42.4 | 62.9 |
| | ChInf | **47.0** | 54.5 | 60.9 | **48.0** | 57.5 | 58.9 | **48.5** | 54.1 | 64.6 |
| MSL | Reconstruct Error | 37.3 | 34.2 | 64.8 | 39.7 | 34.1 | 62.8 | 40.2 | 42.7 | 56.9 |
| | ChInf | **45.8** | 42.2 | 65.4 | **46.2** | 44.6 | 57.1 | **47.7** | 42.8 | 64.9 |
| SWAT | Reconstruct Error | 77.1 | 98.1 | 63.5 | 78.0 | 85.4 | 71.8 | 78.7 | 86.8 | 72.0 |
| | ChInf | **80.1** | 87.7 | 73.7 | **80.6** | 97.6 | 68.6 | **81.9** | 97.7 | 70.6 |
| WADI | Reconstruct Error | 26.7 | 83.4 | 15.9 | 27.5 | 86.2 | 16.3 | 28.9 | 90.8 | 17.2 |
| | ChInf | **44.3** | 84.6 | 30.0 | **46.6** | 83.0 | 32.4 | **47.5** | 71.3 | 35.6 |

## D.4 Additional Dataset and Baseline results

To demonstrate the effectiveness of our approach, we validated our channel-pruning method on new datasets. Additionally, we incorporated a new baseline, DLinear, a time series forecasting method based on a channel-independence strategy. The specific results are shown below:

**New dataset analysis:**

Since the original number of channels in ETTh1 and ETTm1 is only 7, the horizontal axis in the table directly represents the number of retained channels.

Table 9: The additional dataset results of the channel-pruning experiment.

| Dataset | | ETTh1 | | | ETTm1 | | |
|---|---|---|---|---|---|---|---|
| number of channels retained | | 7 | 3 | 2 | 7 | 3 | 2 |
| iTransformer | Continuous selection | 0.396 | 0.502 | 0.573 | 0.332 | 0.756 | 0.826 |
| | Random selection | 0.396 | 0.428 | 0.434 | 0.332 | 0.362 | 0.372 |
| | Influence selection | 0.396 | 0.403 | 0.420 | 0.332 | 0.333 | 0.355 |
| PatchTST | Continuous selection | 0.400 | 0.460 | 0.491 | 0.330 | 0.539 | 0.687 |
| | Random selection | 0.400 | 0.415 | 0.424 | 0.330 | 0.352 | 0.364 |
| | Influence selection | 0.400 | 0.400 | 0.405 | 0.330 | 0.336 | 0.347 |

The results in the table demonstrate the effectiveness of channel pruning based on the ChInf, highlighting that PatchTST and iTransformer exhibit comparable utilization of channel information on the ETTh1 and ETTm1 datasets.

**New forecasting length analysis:**

We have added experimental results for the prediction length of 192. The results are as follows:

Table 10: The 192 forecasting length of the channel-pruning experiment.

| Method | Dataset | ECL | | | | | | Solar | | | | | | Traffic | | | | | |
|---|---|---|---|---|---|---|---|---|---|---|---|---|---|---|---|---|---|---|---|
| | Proportion of variables retained | 5% | 10% | 15% | 20% | 50% | 100% | 5% | 10% | 15% | 20% | 50% | 100% | 5% | 10% | 15% | 20% | 30% | 100% |
| iTransformer | Continuous selection | 0.212 | 0.193 | 0.189 | 0.186 | 0.182 | | 0.270 | 0.260 | 0.256 | 0.251 | 0.249 | | 0.486 | 0.456 | 0.427 | 0.426 | 0.425 | |
| | Random selection | 0.203 | 0.189 | 0.183 | 0.179 | 0.172 | 0.164 | 0.266 | 0.258 | 0.260 | 0.249 | 0.248 | 0.240 | 0.476 | 0.436 | 0.425 | 0.421 | 0.420 | 0.413 |
| | Influence selection | 0.191 | 0.181 | 0.173 | 0.171 | 0.165 | | 0.259 | 0.256 | 0.254 | 0.244 | 0.242 | | 0.460 | 0.430 | 0.422 | 0.416 | 0.413 | |
| | Proportion of variables retained | 5% | 10% | 15% | 20% | 40% | 100% | 5% | 10% | 15% | 20% | 50% | 100% | 5% | 10% | 15% | 20% | 20% | 100% |
| PatchTST | Continuous selection | 0.272 | 0.216 | 0.201 | 0.200 | 0.199 | | 0.282 | 0.270 | 0.265 | 0.264 | 0.260 | | 0.501 | 0.488 | 0.480 | 0.479 | 0.479 | |
| | Random selection | 0.210 | 0.206 | 0.198 | 0.194 | 0.191 | 0.186 | 0.274 | 0.270 | 0.266 | 0.263 | 0.260 | 0.260 | 0.496 | 0.485 | 0.480 | 0.474 | 0.474 | 0.465 |
| | Influence selection | 0.200 | 0.197 | 0.195 | 0.190 | 0.186 | | 0.267 | 0.264 | 0.262 | 0.260 | 0.260 | | 0.485 | 0.475 | 0.470 | 0.465 | 0.465 | |

From the results shown in the table, it can be observed that channel-pruning based on ChInf is more effective. Additionally, iTransformer still exhibits a larger core subset, demonstrating its superior ability to model channel dependency.

**New baseline analysis:**

Table 11: The channel-pruning experiment results of DLinear model.

| Dataset | | ECL | | | | | | Solar | | | | | | Traffic | | | | | |
|---|---|---|---|---|---|---|---|---|---|---|---|---|---|---|---|---|---|---|---|
| Proportion of variables retained | | 5% | 10% | 15% | 20% | 50% | 100% | 5% | 10% | 15% | 20% | 50% | 100% | 5% | 10% | 15% | 20% | 30% | 100% |
| DLinear | Continuous selection | 0.201 | 0.200 | 0.198 | 0.197 | 0.196 | | 0.311 | 0.309 | 0.307 | 0.301 | 0.301 | | 0.649 | 0.647 | 0.645 | 0.645 | 0.645 | |
| | Random selection | 0.200 | 0.198 | 0.196 | 0.196 | 0.196 | 0.196 | 0.306 | 0.304 | 0.303 | 0.301 | 0.301 | 0.301 | 0.649 | 0.648 | 0.645 | 0.645 | 0.645 | 0.645 |
| | Influence selection | 0.197 | 0.196 | 0.196 | 0.196 | 0.196 | | 0.301 | 0.301 | 0.301 | 0.301 | 0.301 | | 0.646 | 0.645 | 0.645 | 0.645 | 0.645 | |

The experimental results in the table show that the core channel subset of DLinear is less than $5\%$, which highlights the limited ability of simple linear models to utilize information from different channels effectively.

### D.5 Difference between Chinf and Channel Pruning Baselines

In our experiment, LIME and SHAP can indeed provide similar channel-wise insights. However, ChInf offers the following unique advantages over these methods:

1.**Computational Efficiency**: Unlike LIME and SHAP, ChInf does not require additional retraining. Once the model is trained, ChInf can be computed directly without the need for perturbation-based retraining (as required by LIME) or an exponential number of model evaluations for different feature combinations (as required by SHAP, which has a complexity of $O(2^n)$ for $n$ features). This makes ChInf significantly more efficient in high-dimensional multivariate time series scenarios.

2.**Beyond Post-hoc Explanations**: To the best of our knowledge, LIME and SHAP serve as post-hoc interpretability tools, meaning they primarily provide insights into feature importance after model training. In contrast, ChInf not only quantifies feature importance but can also be directly leveraged for practical tasks such as anomaly detection, as demonstrated in our paper.

3.**Channel correlation**: While LIME and SHAP are effective interpretability tools, they fail to capture the inter-channel influence during training—a distinctive capability provided by ChInf matrix.

From the table 5, we can observe that when the number of channels is relatively small (e.g., ECL contains 321 channels), LIME and SHAP achieve results similar to ours. However, when the number of channels is large (e.g., Traffic contains 862 channels), these methods perform worse than ours due to their limited approximation capabilities.

Additionally, we have provided a comparison of the time required to compute the weights for different channels in a single MTS on iTransformer. On ECL dataset Chinf : 0.071s, SHAP: 146s LIME:9s.

From the results, it is evident that our method is significantly faster than the other approaches.

### D.6  Additional Complexity analysis results

The computational complexity of ChInf is analyzed as follows: Let $n$ be the number of channels in a multivariate time series, and let $d$ be the number of parameters for which we compute the gradient. The complexity of computing the gradient is $O(d)$, so the overall complexity of ChInf can be expressed as: $O(nd)$ This indicates that the computational cost increases with both the number of channels ($n$) and the parameter dimensionality $d$.

To further illustrate the complexity of our method, we added complexity analysis experiments in both time series anomaly detection and forecasting tasks. In these experiments, we measured the time required to compute the influence of all channels of a single multivariate time series data sample.

**Anomaly detection:**

We have added an experiment measuring the time required for detection at each time point to demonstrate the complexity of our approach, as shown in the table below:

Table 12: The time required for our method on different time series model.

| Dataset | GCN_lstm+ours | iTransformer+ours |
|---------|---------------|-------------------|
| SWAT | 1.4ms | 1.5ms |
| WADI | 6.4ms | 6.5ms |

The results in the table indicate that our detection speed is at the millisecond level, which is acceptable for real-world scenarios.

**Channel-pruning:**

By measuring the time required for calculating single-instance influence, we demonstrated how the computational time scales with the number of channels.

Table 13: The time required for channel-pruning method on different time series datasets.

| | ETTm1 | Solar-Energy | Electricity | traffic |
|---|-------|--------------|-------------|---------|
| iTransformer+ours | 0.0025s | 0.023s | 0.071s | 0.18s |

From the table, it can be observed that the computational complexity approximately increases linearly with the number of channels.

