# OpenReview forum: "Channel Matters: Estimating Channel Influence for Multivariate Time Series"
_NeurIPS.cc/2025/Conference — NeurIPS 2025 poster_

### Official Review · Reviewer_JiPH · 2025-07-01

**Clarity:** 3
**Significance:** 3
**Originality:** 3
**Rating:** 5
**Confidence:** 4

**Summary:**

This paper proposes using influence functions, which estimate the counterfactual effect between training data and model performance, as a data-centric perspective to better understand and analyze multivariate time series (MTS). Addressing the limitation of existing influence functions, such as TimeInf, which overlook the impact of different channels in MTS, the authors introduce a new influence function called ChInf. This method is applied to two key MTS tasks: anomaly detection and data pruning. The results demonstrate that ChInf consistently performs well across various methods, validating its effectiveness.

**Questions:**

1. See Weakness #2 above.
2. In Section 5.1.3, you claim that the original TracIn is ineffective because it doesn't differentiate between channels. This conclusion seems somewhat unconvincing: shouldn't a global influence function still yield meaningful results? According to your reasoning, would TracIn and ChInf perform similarly on univariate time series? Further explanation or empirical validation of this point would be helpful.

**Ethical Concerns:**

["NO or VERY MINOR ethics concerns only"]

**Final Justification:**

I raise the score because all my concerns have been addressed, especially the definition of the final score across different channel.

**Limitations:**

yes

**Quality:**

3

**Strengths And Weaknesses:**

Pros:
1. Applying influence functions to analyze MTS is a novel and meaningful idea.
2. The paper is well-structured and easy to follow, progressing logically from the introduction of influence functions, through their refinement, to the empirical validation of ChInf across multiple tasks.
3. The experiments are thorough, demonstrating ChInf’s effectiveness from multiple perspectives and on various tasks.

Cons:
1. The citation formatting is inconsistent and needs revision.
2. There are concerns about the definition of the influence score in the two downstream tasks:
* For example, in anomaly detection, why is the score defined as the maximum across all channels rather than the average? This choice requires further empirical justification, especially since uniformly distributed influence across channels might be overlooked.
* In Equation (4), why is the score computed only for a channel's influence on itself? Shouldn't channel i potentially influence all channels, not just itself?

---

> ### Author Rebuttal · Authors · 2025-07-31
>
> Thank you for your valuable review and support. We sincerely appreciate your insightful feedback!
>
> **Q1** *The citation formatting is inconsistent and needs revision.*
>
> **A1:** Thank you for pointing this out. We apologize for the inconsistency in citation formatting and will correct all formatting issues in the final version to ensure clarity and professionalism.
>
> **Q2:** *For example, in anomaly detection, why is the score defined as the maximum across all channels rather than the average? This choice requires further empirical justification, especially since uniformly distributed influence across channels might be overlooked.*
>
> **A2:** Thank you for raising this insightful question.
>
> We chose to use the maximum of the self-influence values across channels as the anomaly score to better capture localized, channel-specific anomalies.
>
> To empirically justify this design choice, we highlight the following:
>
> Comparison with TracIn:
> TracIn computes a global influence score by summing over all channels, effectively averaging out the individual channel effects. As a result, it cannot distinguish which specific channels contribute to the anomaly. By comparing our ChInf scores with TracIn’s aggregated results (e.g., Figure 1a/b), we demonstrate the advantage of our fine-grained formulation in capturing localized anomalies. This supports the hypothesis that averaging dilutes anomaly signals, especially when the anomaly is confined to a single or small subset of channels.
>
> Additional datasets for validation:
> We further validated our scoring strategy by adding two widely used MTS anomaly detection benchmarks — SWaT and MSL— in which anomalies are often confined to specific sensors. Our method continues to outperform baselines in these datasets, reinforcing that the maximum-based scoring is more effective for anomaly detection.
>
> |     Model    |  SWaT |     |  MSL  |     |
> |:------------:|:-----:|:---:|:-----:|-----|
> |              | Chinf | TracIn | Chinf | TracIn |
> | iTransformer | 84.0    |  77.1   |  46.1     |   38.3  |
> |   GCN-Lstm   |  82.9   |   76.9  |    47.1   |  40.1   |
>
> Qualitative motivation:
> In the multivariate time series datasets we evaluate on—SWaT, WADI, SMAP, and others—anomalies typically manifest in only a small subset of channels, such as a malfunctioning sensor or a corrupted reading in a single stream. In such settings, aggregating influence scores via mean or sum can dilute the anomaly signal, especially when most channels remain normal. By contrast, using the maximum self-influence across channels effectively surfaces localized anomalies that might otherwise be overshadowed in aggregate-based schemes.
>
> This design choice is empirically supported by our results in Section 5.1.3 and Figure 1a/b, where max-based scoring outperforms global influence baselines (e.g., TracIn) that implicitly average over all channels. While mean- or sum-based scoring might be preferable in scenarios involving uniformly distributed anomalies, such patterns are less common in the evaluated benchmarks. We believe this reflects a broader trend in real-world MTS applications, where localized channel faults are more critical and frequent than global disruptions.
>
> **Q3:** *In Equation (4), why is the score computed only for a channel's influence on itself? Shouldn't channel i potentially influence all channels, not just itself?*
>
> **A3:** Thank you for this thoughtful question.
>
> Our use of self-influence in Equation (4) directly follows the idea of self-influence as introduced by Koh and Liang [1], where the influence of a data point on itself is used as a metric to assess its abnormality or correctness. This formulation has been widely adopted in anomaly detection literature[2] that leverages influence functions.
>
> In the context of multivariate time series, this naturally extends to channel-wise self-influence: the influence of a channel's input on its own loss reflects how much that channel influence the model's behavior, hence indicating potential anomalies. Thus, channel-wise self-influence is enough to detect anomalies. We will clarify this point in the revision to avoid confusion.
>
>
>
> **Q4:** *In Section 5.1.3, you claim that the original TracIn is ineffective because it doesn't differentiate between channels. This conclusion seems somewhat unconvincing: shouldn't a global influence function still yield meaningful results? According to your reasoning, would TracIn and ChInf perform similarly on univariate time series? Further explanation or empirical validation of this point would be helpful.*
>
> **A4:** Thank you for this insightful observation.
>
> We would like to clarify that we do not claim TracIn to be entirely ineffective, while it is less effective than the proposed Chinf. Rather, our intention is to point out that TracIn operates at the sample level and does not distinguish between individual channels. Consequently, it may overlook localized anomalies that are specific to certain channels. In comparison, per-channel reconstruction losses of the base model serve as a proxy for identifying local deviations. We acknowledge that our previous wording may have caused confusion, and we are happy to revise the text to better reflect this distinction.
>
> On univariate time series, TracIn and ChInf reduce to the same form — we fully agree. In fact, ChInf is designed to generalize TracIn to the multivariate setting, offering better resolution in MTS scenarios.
>
> In multivariate settings, TracIn produces one score per sample (treating all channels together), while ChInf computes a channel-wise matrix, offering significantly richer information. This is particularly helpful for tasks like channel pruning, where individual channel importance matters.
>
> Empirically, Figure 1a and 1b show that TracIn performs worse than ChInf. This supports our claim that channel-wise resolution is essential in MTS, while TracIn is not sufficient.
>
>
> [1] Understanding Black-box Predictions via Influence Functions ICML 2017
>
> [2] Time Series Data Contribution via Influence Functions ICLR 2025
>
> Finally, we sincerely thank the reviewer’s feedback. It definitely inspires us to further improve our work and clarification for more readers. We respectfully hope that the reviewer could reevaluate our work given the responses addressing your main concerns.

---

> > ### Comment · Reviewer_JiPH · 2025-08-04
> > **response to the rebuttal**
> >
> > Thank you for the author’s rebuttal. My concerns have been addressed, and I will raise the score accordingly.

---

> > > ### Author Response · Authors · 2025-08-04
> > >
> > > Thank you very much for your thoughtful response and for updating your score. We truly appreciate your recognition of our efforts in addressing your concerns. Your constructive feedback has been invaluable in improving our work, and we are grateful for your support.

---

### Official Review · Reviewer_g6Y3 · 2025-07-03

**Clarity:** 3
**Significance:** 3
**Originality:** 2
**Rating:** 5
**Confidence:** 3

**Summary:**

The paper introduces a method for quantifying how removing a channel from a multivariate time-series affects the loss on a another channel. This metric is applied to standard time-series tasks including anomaly detection and data pruning.

**Questions:**

Re-iterating my previous questions:
- Why do simple baseline methods outperform most deep methods, and what differentiates ChInf? Even if that is mentioned in a cited work, a sentence explanation would be helpful to aid in interpretation of the results.
- What kind of anomalies do you expect ChInf to be strong at detecting and where do you expect it to perform less favorably. Is there something about the tested datasets that indicates that they are well suited for ChInf? Are there cases where you expect the proposed method to underperform? Some additional discussion of this point would be helpful.

**Ethical Concerns:**

["NO or VERY MINOR ethics concerns only"]

**Final Justification:**

I have increased my score. The authors have addressed my questions and I think their revisions make their contribution more clear.

**Limitations:**

See my question regarding anomalies that may be "missed" by this method.

**Paper Formatting Concerns:**

No major concerns but please proof read the submission for typos and grammatical errors, e.g. "While channel extremely matters" in the abstract.

**Quality:**

3

**Strengths And Weaknesses:**

The paper is generally well written and the method seems sound albeit somewhat incremental. In the context of anomaly detection, it seems to amount to a different aggregation of per-channel influence (max instead of sum) if I understand correctly. Still the method seems useful in practice. The empirical analysis is well done and includes simple baselines that seem to outperform many competing deep models. Some discussion of this results would be helpful, along with why the proposed method is stronger. This leads me to a larger point:

A common view regarding anomaly detection is that some prior knowledge is needed regarding the down-stream task in order to define what constitutes an anomaly [1,2]. This makes evaluation and comparison of anomaly detection methods challenging. I think this work could be stronger if there was at least a discussion on the *kinds* of anomalies the method targets. For instance it would seem to me that the method should be strong at detecting anomalies when individual channels are "broken", meaning they do not follow their usual pattern in relation to other channels (for instance, this work [3] tackles such anomalies in games by modeling correlations between channels). This might be in contrast to methods that are better at detecting when the entire time-series behaves abnormaly.

I would also recommend that the authors revise some of the method explanation, for instance after eq. 1, to improve clarity.

[1] Winkens, Jim, et al. "Contrastive training for improved out-of-distribution detection." arXiv preprint arXiv:2007.05566 (2020).
[2] Le Lan, Charline, and Laurent Dinh. "Perfect density models cannot guarantee anomaly detection." Entropy 23.12 (2021): 1690.
[3] Lymperopoulos, Panagiotis, Yukun Li, and Liping Liu. "Exploiting variable correlation with masked modeling for anomaly detection in time series." NeurIPS 2022 Workshop on Robustness in Sequence Modeling. 2022.

---

> ### Author Rebuttal · Authors · 2025-07-31
>
> Thank you for your valuable review and support. We sincerely appreciate your insightful feedback!
>
> **Q1:** *I would also recommend that the authors revise some of the method explanation, for instance after eq. 1, to improve clarity.*
>
> **A1:** Thank you for the suggestion. We will revise the explanation after Equation (1) to better highlight the distinction between sample-level and channel-level influence computation, and clarify how the Chinf matrix term reflects the structural interactions among channels. We have revised the paragraph to enhance readability and make the derivation and intuition behind TracIn and our proposed ChnInf more accessible to readers. The revised version is as follows:
>
> > **[Modified]**
> > "The influence function [Koh and Liang, 2017] estimates the effect of removing a training example on the model’s performance.
> However, its exact computation requires inverting a Hessian matrix, which is computationally expensive. TracIn [Pruthi et al., 2020] proposes an efficient approximation by measuring the change in the loss of a test point as a model is trained with a specific training point. Formally, the influence of training sample \( z' \) on test sample \( z \) is approximated as:
> > \[
> > $\text{TracIn}(z', z) = L(z; \theta) - L(z; \theta') \approx \eta \nabla_\theta L(z'; \theta)^\top \nabla_\theta L(z; \theta)$
> > \]
> > where \( z' \) is a training example, \( z \) is a test example, \( \theta \) is the model parameter before training on \( z' \), \( \theta' \) is the updated parameter after training on \( z' \), and \( \eta \) is the learning rate. This formulation approximates the influence of \( z' \) by the inner product of the gradients of the training and test losses, scaled by \( \eta \)."
>
> >However, in multivariate time series (MTS) scenarios, both $z$ and $z'$ represent MTS data. TracIn computes the overall influence of the entire input but cannot distinguish how individual channels contribute. This limitation makes it difficult to analyze channel-wise influence. To address this, we propose a new method, \textbf{ChnInf}, that decomposes TracIn-style influence estimation into per-channel contributions. Our derivation, which follows a similar principle to TracIn, is presented in Theorem 3.1, with full proof provided in Appendix B.
>
> We hope this revision improves the clarity of our method section. We have also ensured that similar improvements are made throughout the revised manuscript.
>
>
>
>
>
> **Q2：** *Why do simple baseline methods outperform most deep methods, and what differentiates ChInf? Even if that is mentioned in a cited work, a sentence explanation would be helpful to aid in interpretation of the results.*
>
> **A2:** We thank the reviewer for raising this important point. As noted in prior work [1], simple models often outperform deep architectures in MTS anomaly detection because normal patterns in these datasets are relatively regular and can be captured by shallow models. The added complexity of deep models provides limited marginal gain, consistent with the principle of Ockham’s Razor. Moreover, as discussed in [1], previous deep methods sometimes showed inflated performance due to flawed evaluation protocols, such as using 'point adjustment'.
>
> ChInf differs fundamentally from both shallow and deep detection methods. First, ChInf is model-agnostic—it can be applied on top of any backbone model without modifying the architecture or retraining. Second, ChInf leverages gradient-based self-influence scores, which provide a model-centric perspective for anomaly detection by quantifying how much a data point (or channel) influences its own loss landscape. This diagnostic signal reflects how sensitive the model’s parameters are to perturbations in specific inputs, thus capturing subtle anomalies that may not result in large prediction errors.
>
> **Q3:** *What kind of anomalies do you expect ChInf to be strong at detecting and where do you expect it to perform less favorably. Is there something about the tested datasets that indicates that they are well suited for ChInf? Are there cases where you expect the proposed method to underperform? Some additional discussion of this point would be helpful.*
>
> **A3:** We appreciate the reviewer’s insightful question. ChInf is particularly effective at detecting localized, channel-wise anomalies, where individual channels deviate from their typical interactions with others. These cases often result in unexpected gradients, indicating that the model’s parameters are unusually sensitive to the input—a scenario naturally captured by self-influence.
>
> This aligns with the reviewer’s suggestion regarding “broken channels”, and is corroborated by findings in the references which you list, which highlight the value of modeling channel dependencies.
>
> Conversely, ChInf may be less effective in settings, where anomalies are purely temporal across all channels.
>
> As for the datasets used in our experiments, they often contain small subset-sensor failures or spikes, making them naturally aligned with ChInf’s strengths. We will add previous discussion and cite the references you provided to the Appendix to further clarify where ChInf’s advantages lie.
>
> [1] Position Paper: Quo Vadis, Unsupervised Time Series Anomaly Detection?
>
> Finally, we sincerely thank the reviewer’s feedback. It definitely inspires us to further improve our work and clarification for more readers. We respectfully hope that the reviewer could reevaluate our work given the responses addresing your main concerns.

---

> > ### Comment · Reviewer_g6Y3 · 2025-08-03
> >
> > Thank you for your response. I have raised my score based on your improved explanations

---

> > > ### Author Response · Authors · 2025-08-03
> > > **Response to Reviewer**
> > >
> > > Thank you very much for your thoughtful response and for updating your score. We truly appreciate your recognition of our efforts in addressing your concerns. Your constructive feedback has been invaluable in improving our work, and we are grateful for your support.

---

### Official Review · Reviewer_MH5D · 2025-07-03

**Clarity:** 3
**Significance:** 3
**Originality:** 3
**Rating:** 5
**Confidence:** 2

**Summary:**

This work proposes Channel-wise Influence (ChInf), a novel method that extends influence functions to multivariate time series (MTS) data by estimating the influence of individual channels. While influence functions have traditionally been used to analyze the effect of training data on model parameters without retraining, their application to channel-centric analysis in MTS tasks has been largely unexplored. To address this gap, the authors introduce ChInf and derive two ChInf-based algorithms for common MTS tasks. Experimental results show that these methods significantly outperform existing approaches in MTS anomaly detection and data pruning, highlighting the necessity and effectiveness of channel-wise influence analysis.

**Questions:**

1. Could you provide additional experiments on MTS anomaly detection using the UEA archives, as prior works on MTS anomaly detection typically include evaluations on these datasets?

2. Can you provide more theoretical analyses on this work? For instance, the computational complexity?

3. It's clear that the influence of MTS in deep learning has not been well studied, and how to estimate the influence of different channels in MTS is critical. However, what are the challenges in the process remains unclear. More discussions are required.

4. The authors are suggested to add more discussions into the related work. For instance, the discussions on the MTS anomaly detection and forecasting are limited. Additional discussions on the categories of MTS anomaly detection or forecasting can enhance the readability of this work.

**Ethical Concerns:**

["NO or VERY MINOR ethics concerns only"]

**Final Justification:**

The authors have addressed my concerns.

**Limitations:**

Yes

**Quality:**

3

**Strengths And Weaknesses:**

Strengths:

1. This study is interesting, which investigate the influence functions to multivariate time series (MTS) data by estimating the influence of individual channels.

2. The writing and organization of this paper are good.

3. The proposed method shows excellent performance.

However, I have some concerns as follows.

Weaknesses:

1. Could you provide additional experiments on MTS anomaly detection using the UEA archives, as prior works on MTS anomaly detection typically include evaluations on these datasets?

2. Can you provide more theoretical analyses on this work? For instance, the computational complexity?

3. It's clear that the influence of MTS in deep learning has not been well studied, and how to estimate the influence of different channels in MTS is critical. However, what are the challenges in the process remains unclear. More discussions are required.

4. The authors are suggested to add more discussions into the related work. For instance, the discussions on the MTS anomaly detection and forecasting are limited. Additional discussions on the categories of MTS anomaly detection or forecasting can enhance the readability of this work.

---

> ### Author Rebuttal · Authors · 2025-07-31
>
> Thank you for your valuable review and support. We sincerely appreciate your insightful feedback!
>
> **Q1：** *Could you provide additional experiments on MTS anomaly detection using the UEA archives, as prior works on MTS anomaly detection typically include evaluations on these datasets?*
>
> **A1:** Thank you for your valuable comment. To the best of our knowledge, the UEA time series archive primarily consists of datasets designed for classification tasks and does not include benchmarks specifically curated for multivariate time series  anomaly detection. We carefully reviewed the archive and found no prior works that utilize UEA datasets for MTS anomaly detection.
>
> If your comment refers instead to the UCR archive, we would like to clarify that UCR datasets are univariate.  Since our method is designed to analyze channel-wise influence in multivariate settings, univariate datasets such as those in the UCR archive fall outside the scope of our study.  We thus focus our evaluation on multivariate time series benchmarks.
>
> If we have misunderstood your suggestion, we would greatly appreciate it if you could clarify your intent and, if possible, point us to prior work that uses UEA datasets for anomaly detection. We would be happy to include those references or consider additional experiments accordingly.
>
> **Q2:** *Can you provide more theoretical analyses on this work? For instance, the computational complexity?*
>
> **A2:** Thank you for the suggestion. We agree that theoretical insights can strengthen our contribution. We have introduced the complexity analysis in Appendix D.6. The details are as follows:
>
> ChInf is based on first-order gradient approximations, and thus the per-sample complexity is linear in the number of channels N and linear in the number of parameters used for influence estimation (which can be limited to the final layer, as shown in Fig. 1c). Specifically, the complexity is: O(N⋅d)
>
> where d is the number of parameters chosen (e.g., final-layer parameters), and N is the number of channels. In practice, we show (Fig. 1c) that using only a small subset of parameters still yields strong performance, making ChInf highly scalable.
>
>
> **Q3:** *It's clear that the influence of MTS in deep learning has not been well studied, and how to estimate the influence of different channels in MTS is critical. However, what are the challenges in the process remains unclear. More discussions are required.*
>
> **A3:** Thank you for raising this important conceptual question. Indeed, one of the central challenges we address in this work is extending influence estimation from the conventional sample level to the channel level in multivariate time series.
>
> While this shift might appear straightforward at first glance, it poses a fundamental theoretical and algorithmic challenge: standard influence functions are intrinsically designed for sample-wise analysis, and cannot be directly applied to assess the impact of individual channels. This is because in most formulations, influence is computed by perturbing or removing an entire training sample — a complete MTS instance — and evaluating its effect on the test loss.
>
> In contrast, estimating the channel-wise influence requires a rethinking of the foundational assumptions. Specifically, we must redefine what it means to "perturb" only a single channel, while preserving the temporal and structural integrity of the time series. This entails a new formulation that decouples the loss sensitivity with respect to each channel's contribution, which is non-trivial not only mathematically and computationally, but also conceptually.
>
> Our work provides such a formulation, along with theoretical justification and practical approximations. In this sense, the main challenge lies in conceptualizing and enabling channel-wise influence estimation as a first-class tool, something prior influence-function-based works in vision or NLP have not addressed.
>
> **Q4:** *The authors are suggested to add more discussions into the related work. For instance, the discussions on the MTS anomaly detection and forecasting are limited. Additional discussions on the categories of MTS anomaly detection or forecasting can enhance the readability of this work.*
>
> **A4:** We appreciate the reviewer’s suggestion. We will revise the related work section to include a more comprehensive discussion of MTS anomaly detection and forecasting.
>
> MTS Anomaly Detection:
> Existing MTS anomaly detection methods can be broadly categorized into two groups. The first group consists of model-centric methods, which estimate anomaly scores via prediction or reconstruction errors (e.g., TransAD,USAD, GDN, GCN-LSTM and so on). Within this group, certain models explicitly model channel dependencies. For example, GDN and GCN-LSTM use graph convolutions to represent the structural relationships among sensors and capture correlations across channels. The second group includes data-centric methods, which leverage influence measures to evaluate the impact of inputs (e.g., TimeInf, our proposed ChInf). These methods do not rely on explicit temporal or graph modeling, but instead use influence-based diagnostics to reflect how sensitive the model is to input perturbations. We emphasize that our work belongs to the latter category and is one of the first to provide channel-wise influence analysis, offering better granularity and interpretability for channel-level anomaly detection.
>
> MTS Forecasting:
> For multivariate time series forecasting, recent studies show that the training paradigm can significantly affect model performance. There are two typical training paradiam. A number of works adopt channel-dependent models (e.g., iTransformer), which aim to capture channel dependencies by jointly modeling all channels. Others use channel-independent models (e.g., PatchTST), which make the parameter of the model shared across all channels to avoid inter-channel interference. Our proposed ChInf can be integrated into either type of forecasting model to analyze the influence of individual channels, allowing deeper insight into channel contributions regardless of the model’s training paradigm.
>
> Finally, we sincerely thank the reviewer’s feedback. It definitely inspires us to further improve our work and clarification for more readers. We respectfully hope that the reviewer could reevaluate our work given the responses addresing your main concerns.

---

> > ### Comment · Reviewer_MH5D · 2025-08-03
> > **Thanks for the rebuttal**
> >
> > Thanks for the rebuttal. Most of my concerns have been addressed. For the prior works that utilize UEA datasets for MTS anomaly detection, you can find such discussions about how to build the training, validation, and testing sets in [1, 2]. As a reviewer, I understand it's hard for the authors to provide some experimental results during the rebuttal period. Thanks.
> >
> > [1] Unsupervised outlier detection for time series by entropy and dynamic time warping
> >
> > [2] TimeAutoAD: Autonomous Anomaly Detection With Self-Supervised Contrastive Loss for Multivariate Time Series.

---

> > > ### Author Response · Authors · 2025-08-03
> > > **Response to Reviewer**
> > >
> > > Thank you very much for the reference you provided. We have carefully read the paper. If we understand correctly, the anomaly detection dataset used in that work is derived from a classification dataset. According to the construction rules described in the paper, the anomalous samples are typically at the sample level, as they belong to entirely different classes with completely different distributions across channels. This setting falls outside the scope of our work. For a more detailed discussion on the types of datasets our method is particularly suited for, we kindly refer you to our response to Reviewer g6Y3’s Q3, where we elaborated on this aspect.
> > > Moreover, we would sincerely appreciate it if you would consider raising your score, as most of your main concerns have been solved. Your suggestion and action would both contribute positively to the research community.

---

> ### Comment · Reviewer_MH5D · 2025-08-03
>
> Thanks for the detailed explanations. I will raise the score accordingly. Besides, do you think if it will benifit this manuscript if you add more discussions in the Limitation section about why the proposed method can not be applied to this scenario?

---

> ### Author Response · Authors · 2025-08-03
> **Response to Reviewer**
>
> Thank you very much for your thoughtful response and for updating your score. Thank you very much for your constructive comments. We would be happy to incorporate the type of datasets referenced in the papers you mentioned into our discussion—particularly by connecting them with the points raised by Reviewer g6Y3. We plan to integrate this into the Limitation section or other appropriate parts of the paper to further clarify the scope of our method and help readers better understand its positioning.

---

### Official Review · Reviewer_EHTb · 2025-07-07

**Clarity:** 2
**Significance:** 3
**Originality:** 2
**Rating:** 5
**Confidence:** 3

**Summary:**

This paper proposes Channel-wise Influence (ChInf), a novel influence function for Multivariate Time Series (MTS) data, which estimates the influence of each channel, allowing both interpretability and efficient channel pruning. The authors validate their method through experiments on anomaly detection and channel pruning, demonstrating its effectiveness, interpretability, and generality across multiple datasets and model architectures (e.g., iTransformer, PatchTST).

**Questions:**

1. For online anomaly detection, is it feasible to update ChInf incrementally?
2. In the context of real-world datasets such as Electricity, is there any case study, qualitative analysis or interpretability experiment demonstrating whether the channels ranked by ChInf correspond to domain-critical variables or sensors? Do these rankings align with known domain knowledge from the respective domains?

**Ethical Concerns:**

["NO or VERY MINOR ethics concerns only"]

**Final Justification:**

The authors have addressed my concerns adequately.

**Limitations:**

Yes

**Quality:**

3

**Strengths And Weaknesses:**

Strengths:
1. This paper introduces the Channel-wise Influence Function, a new method designed to assess and interpret the influence relationships among different channels in MTS data. In contrast, the traditional methods assess the impact of entire data samples on model performance.
2. ChInf proves to be a versatile approach, achieving good performance in two MTS tasks—anomaly detection and channel pruning.

Weakness:
1. The method requires computing gradients individually for each channel, leading to a computational complexity that scales linearly with the number of channels, which may pose challenges when applied to large-scale multivariate time series data.
2. The method is evaluated on two representative tasks, without investigating its applicability to other important MTS problems, which may limit the demonstration of its broader generalizability.

---

> ### Author Rebuttal · Authors · 2025-07-31
>
> Thank you for your valuable review and support. We sincerely appreciate your insightful feedback!
>
>
> **Q1** *The method requires computing gradients individually for each channel, leading to a computational complexity that scales linearly with the number of channels, which may pose challenges when applied to large-scale multivariate time series data.*
>
> **A1:** We appreciate the reviewer’s observation. We acknowledge that the computational complexity of ChInf scales linearly with the number of channels, since it requires computing per-channel gradients. However, in practice, we find that this cost is manageable for two reasons:
> (1) MTS models used in practice are typically lightweight, especially in industrial or real-time settings; (2) ChInf only requires partial gradient computation with respect to the input rather than full backpropagation.
>
> As a result, the actual computation time per sample remains in the order of milliseconds for most benchmark datasets with dozens of channels, suggesting that the computational cost is not a major bottleneck in practice. We provide an empirical analysis in Figure 1(c) of the main paper and further report implementation details in Appendix D.6. While ChInf requires channel-wise gradient computation, our experiments show it remains practical for common MTS settings. Moreover, improving the efficiency of influence estimation—such as through approximation techniques or model-specific acceleration—remains a promising direction for future work.
>
> **Q2** *The method is evaluated on two representative tasks, without investigating its applicability to other important MTS problems, which may limit the demonstration of its broader generalizability.*
>
> **A2:** We thank the reviewer for highlighting this point.     We chose to evaluate ChInf on anomaly detection and time series forecasting because they are among the most widely studied and practically relevant tasks in multivariate time series research.     Moreover, these two tasks correspond to classical use cases of influence functions in machine learning: anomaly detection relates to self-influence, while forecasting and pruning relate to data selection and input attribution.     We believe this aligns with the foundational motivations behind influence-based analysis.
>
> That said, ChInf is a general-purpose tool and is not limited to these tasks.     One promising direction we have begun to explore is using ChInf to reweight the training process.     Specifically, at the end of each training epoch, one could compute the ChInf matrix to assess inter-channel importance, and use it to adjust sample or channel weights in the next epoch.
>
> Due to time constraints, we could not include extensive experiments on this training-time integration in the current version, but we believe it is a valuable extension that further showcases the generality and practicality of ChInf.     We plan to include this direction in future work.
>
> **Q3：** *Questions:For online anomaly detection, is it feasible to update ChInf incrementally?*
>
> **A3:** Thank you for this interesting question. If we understand the reviewer’s suggestion correctly, it refers to updating ChInf in an online manner, where the influence scores are recomputed incrementally as the model updates over time. We believe this is indeed feasible. Since ChInf relies on gradients with respect to the current model parameters, as long as the model is continuously updated (e.g., via streaming or online learning), the self-influence scores can be recomputed at each time step using the latest weights, thereby enabling real-time or online influence tracking.
>
> While we have not explicitly implemented this setting in our current work, we consider it a promising extension. If we have misunderstood the question, we kindly invite the reviewer to clarify, and we would be happy to further elaborate.
>
> **Q4:** *In the context of real-world datasets such as Electricity, is there any case study, qualitative analysis or interpretability experiment demonstrating whether the channels ranked by ChInf correspond to domain-critical variables or sensors? Do these rankings align with known domain knowledge from the respective domains?*
>
>
> **A4:** We thank the reviewer for raising this valuable point regarding the alignment between ChInf-based rankings and known domain-critical variables. While the Electricity dataset contains a large number of channels, making qualitative interpretation difficult, we conducted a focused case study on the ETTh1 dataset, which contains 7 channels with clear physical meanings:
> HUFL(0) (High Utilization Factor Load), HULL(1) (High Utilization Low Load), MUFL(2)(Medium Utilization Factor Load), MULL(3)(Medium Utilization Low Load), LUFL(4)(Low Utilization Factor Load), LULL(5)(Low Utilization Low Load), and OT(6) (Oil Temperature).
>
> According to our experiment, ChInf-based pruning retains high forecasting accuracy when reducing the input to just 4 channels.
>
> Based on our ChInf rankings, when retaining 4 channels, our method selects HULL (1), MUFL (2), LUFL(4), and OT(6). These selected subsets yield performance comparable to using all channels.
>
> We further analyzed the cross-channel relationships using dataset visualization experiment. The patterns show that:
>
>     HUFL (0) and MUFL (2) exhibit similar temporal dynamics.
>     HULL (1) and MULL (3) exhibit similar temporal dynamics.
>     LUFL (4) and LULL (5) present relatively weak similarity.
>     OT (6) shows little correlation with other channels.
>
> These findings suggest that ChInf effectively identifies representative channels, retaining diversity. The selected channels (e.g., 1, 2, 4, 6) represent distinct temporal behaviors while covering correlated patterns, aligning with domain intuition.
>
> Due to rebuttal constraints, we cannot modify the appendix or include figures, but we plan to provide qualitative visualizations and a more detailed interpretability analysis in the final version. These findings suggest that ChInf rankings do reflect meaningful channel importance aligned with domain structure.
>
> Finally, we sincerely thank the reviewer’s feedback. It definitely inspires us to further improve our work and clarification for more readers. We respectfully hope that the reviewer could reevaluate our work given the responses addresing your main concerns.

---

> > ### Comment · Reviewer_EHTb · 2025-08-09
> >
> > Thank you for the author’s rebuttal. As some of my concerns have been resolved and the analyses are thorough, I am willing to increase my score accordingly.

---

> > > ### Author Response · Authors · 2025-08-09
> > >
> > > Thank you very much for your thoughtful response and for updating your score. We truly appreciate your recognition of our efforts in addressing your concerns. Your constructive feedback has been invaluable in improving our work, and we are grateful for your support.

---

### Decision · Program_Chairs · 2025-09-17

**Decision:**

Accept (poster)

**Comment:**

The authors introduce novel channel-wise influence functions that measure counterfactual effects. They evaluate their method for channel pruning and anomaly detection and show that the contributions clearly improve the results.

The reviewers highlighted the novelty of the approach and that the experiments show clear advantages of introducing the channel wise formulation. Criticism included lack of clarity in certain parts and missing theoretical analysis, which the authors could partially provide during the discussion phase. All in all, the authors convinced all reviewers to converge to accept recommendations, which I intend to follow.

After checking the paper myself and discussing with the SAC, we noticed that the paper lacks detailed experimental descriptions, e.g. if and how validation splits were used, or how individual hyperparameters have been chosen. In order to enable reproduction of the work, we heavily encourage to expand the description of experimental setup for the camera ready version.